# Macrophage Activation Markers Predict Liver-Related Complications in Primary Biliary Cholangitis

**DOI:** 10.3390/ijms23179814

**Published:** 2022-08-29

**Authors:** Yukihisa Fujinaga, Tadashi Namisaki, Yuki Tsuji, Junya Suzuki, Koji Murata, Soichi Takeda, Hiroaki Takaya, Takashi Inoue, Ryuichi Noguchi, Yuki Fujimoto, Masahide Enomoto, Norihisa Nishimura, Koh Kitagawa, Kosuke Kaji, Hideto Kawaratani, Takemi Akahane, Akira Mitoro, Hitoshi Yoshiji

**Affiliations:** 1Department of Gastroenterology, Nara Medical University, 840 Shijo-cho, Kashihara 634-8522, Nara, Japan; 2Department of Evidence-Based Medicine, Nara Medical University, 840 Shijo-cho, Kashihara 634-8522, Nara, Japan

**Keywords:** soluble CD163, soluble mannose receptor, primary biliary cholangitis, intestinal permeability markers

## Abstract

Primary biliary cholangitis (PBC) has a wide variation in clinical presentation and course. There is no significant correlation between these symptoms and the disease stage, although patients with more advanced stages generally have more symptoms. It is important to develop biomarkers in order to identify patients with an increased risk of complications and end-stage liver disease. This study investigated surrogate markers for risk estimation of PBC-related complications, including a study population of 77 patients with PBC who underwent liver biopsy and were measured for serum levels of macrophage activation markers, soluble CD163 (sCD163), soluble mannose receptor (sMR), and zonulin. Patients with PBC were divided into symptomatic (Group S, *n* = 20) and asymptomatic (Group A, *n* = 57) groups. The correlations of histological stages based on both Scheuer and Nakanuma classifications with the three serum markers were investigated. The Nakanuma classification involves grading for liver fibrosis and bile duct loss. The three biomarkers were assessed for their diagnostic ability to identify patients with PBC having high risk of developing complications. The predictive factors of these complications were examined as well. Group S had significantly higher serum sMR (*p =* 0.011) and sCD163 (*p =* 0.048) levels versus Group A. A composite index of sMR and sCD163 measurements had significantly better prediction performance than sCD163 alone (*p =* 0.012), although not when compared to sMR alone (*p =* 0.129). Serum sMR was an independent factor for developing complications on both univariate (Odds ratio (OR) = 30.20, 95% confidence interval (95% CI): 3.410–267.0, *p =* 0.00220), and multivariate (OR = 33.70, 95% CI: 3.6600–311.0, *p =* 0.0019) analyses. Patients with PBC having sMR of ≥56.6 had a higher incidence of clinical complications versus those with a sMR of <56.6. Serum sMR predicts the development of complications in patients with PBC. sMR plus sCD163 showed better predictive power than either marker alone, although the addition of sCD163 did not improve the predictive power of sMR. Future prospective studies are required in order to validate the findings of the present study.

## 1. Introduction

Primary biliary cholangitis (PBC) is characterized by immune-mediated injury of intrahepatic bile ducts and progressive fibrosis that eventually result in liver failure [1]. Notably, patients with early-stage PBC treated with ursodeoxycholic acid (UDCA) have comparable prognosis with age-matched healthy controls [2]. Hence, more patients with early-stage PBC are now being treated with UDCA [3]; this emphasizes the importance of having surrogate endpoints for the identification of those who may benefit from additional treatment [4,5]. Approximately 80% of asymptomatic patients have a higher risk of becoming symptomatic within ten years [6]. However, at baseline, only 20% are symptomatic [7]. The American Association for the Study of Liver Diseases Practice Guidance on PBC found no significant correlation between these symptoms and disease stage, although patients with more advanced disease stages generally have more symptoms [8] Among twenty (26.0%) out of the seventy-seven patients overall, twelve developed pruritus, two ascites, six esophageal varices, two hepatocellular carcinoma, and two jaundice, including patients who developed several symptoms. Fifty-seven (84.0%) patients remained symptomatic throughout the study period. Nevertheless, PBC has a wide variation in clinical presentation and course. The most common symptom of PBC requiring treatment in patients with PBC is pruritus [9]. Pruritus severity is unrelated to the severity of the liver disease. Therefore, it is important to develop biomarkers in order to identify such patients who have an increased risk of complications and end-stage liver disease.

Disturbances of the gut–liver axis are associated with various chronic liver diseases [10]. Epithelial barrier function and innate immunity both play crucial roles in the pathogenesis of PBC and its various forms of autoimmunity [7]. Having a defective epithelial barrier and increased intestinal permeability (IP) can promote an immunological response and elevate levels of circulating endotoxin [11]. Altered intestinal bacteria profiles and increased IP have been observed in patients with PBC [12]. The macrophage activation markers soluble CD163 (sCD163) and mannose receptor (sMR) correlate with the lactulose mannitol ratio, which is a widely used method of assessing IP in patients with cirrhosis [13]. Moreover, sCD163 and sMR are known markers of liver disease severity and prognosis in patients with PBC [14], although there is a need to explore their use in assessing complications and end-stage liver disease. In this study, we aimed to identify biomarkers to predict the development of clinical symptoms in patients with PBC.

## 2. Results and Discussion

### 2.1. Baseline Clinical Characteristics of Patients

Table 1 illustrates a summary of the demographics and baseline characteristics of the 77 patients. Among them, 66 (85.7%) were women, and the median age at diagnosis was 63.5 ± 9.8 years. Out of 77 patients, 20 (26.0%) developed pruritus (*n* = 12), ascites (*n* = 2), esophageal varices (*n* = 6), hepatocellular carcinoma (HCC) (*n* = 2), and jaundice (*n* = 2) during the study period. Moreover, three patients died of liver-related events, two had HCC, and one had liver failure. The distribution of patients in stage I, II, III, and IV based on the Scheuer classification was 25 (32.5%), 39 (50.6%), 11 (14.3%), and 2 (2.6%), respectively. On the other hand, the distribution of patients in stage 1, 2, 3, and 4 based on the Nakanuma classification was 6 (7.8%), 28 (36.4%), 40 (51.9%), and 3 (3.9%), respectively. Fibrosis scores of 0, 1, 2, and 3 were seen in 19 (24.7%), 43 (55.8%), 13 (16.9%), and 2 (2.6%) patients, respectively. BDL scores of 0, 1, 2, and 3 were seen in 7 (9.1%), 32 (41.6%), 24 (31.2%), and 14 (18.2%) patients, respectively. The mean follow-up period was 4.4 ± 1.8 years. In symptomatic patients, serum ALB levels were significantly lower (*p =* 0.026), whereas serum levels of sMR (*p =* 0.011) and sCD163 (*p =* 0.048) were significantly higher compared to asymptomatic patients.

### 2.2. Correlation of Histological Stage with IP Markers

Serum sMR levels were significantly correlated with Nakanuma fibrosis score (R = 0.36, *p =* 0.019), but not with Scheuer stage, Nakanuma stage, or Nakanuma bile duct score (R = 0.16, *p =* 0.928; R = 0.13, *p =* 0.251; and R = 0.08, *p =* 0.626, respectively) (Figure 1). Serum sCD163 levels were significantly correlated with Nakanuma stage (R = 0.26, *p =* 0.0229), Nakanuma fibrosis score (R = 0.227, *p =* 0.0468), and BDL score (R = 0.27, *p =* 0.0171), but not with Scheuer stage (R = 0.13, *p =* 0.507) (Figure 2). Serum zonulin levels were correlated with Scheuer stage (R = 0.42, *p =* 0.0054), Nakanuma stage (R = 0.33, *p =* 0.0083), and Nakanuma fibrosis score (R = 0.45, *p =* 0.0027), but not with Nakanuma bile duct score (R = 0.16, *p =* 0.0042) (Figure 3).

### 2.3. Diagnostic Accuracy of IP Markers for Symptom Development in Patients with Primary Biliary Cholangitis

The individual cutoff values for IP markers for predicting the development of PBC-related complications, as well as their sensitivity, specificity, positive predictive value, and negative predictive value, are shown in Table 2. An sMR cutoff of 30.1 provided the greatest diagnostic accuracy (area under the receiver operating characteristics curve (AUROC): 0.666). The predictive accuracies of sCD163, sMR, a composite index based on s163, and sMR for symptom development were 59.6% (35.0% sensitivity, 84.2% specificity), 66.6% (35.0% sensitivity, 98.2% specificity), and 73.9% (60% sensitivity, 82.5% specificity), respectively. The combined model had significantly better prediction performance than sCD163 (*p =* 0.012), although not when compared to sMR (*p =* 0.129) (Figure 4).

### 2.4. Predictive Factors for PBC-Related Complications

Serum sMR was an independent factor for developing complications on univariate (Odds ratio (OR) = 30.20, 95% confidence interval (95% CI): 3.410–267.0, *p* = 0.00220), and multivariate (OR = 33.70, 95% CI: 3.6600–311.0, *p* = 0.0019) analysis (Table 3).

### 2.5. Cumulative Incidence of Clinical Symptom in PBC

The cumulative incidence of PBC-related complications according to sMR and sCD163 are shown in Figure 5 and Figure 6, respectively. Patients with a sMR of ≥56.6 had a significantly higher incidence of clinical complications than those with an sMR of <56.6 (*p* < 0.001). On the other hand, those with an sCD163 of ≥30.1 tended to have a higher incidence of clinical complications than those with a CD163 of <30.1, although this was not statistically significant (*p =* 0.0552).

## 3. Materials and Methods

### 3.1. Patient Population

This was a prospective observational study that enrolled a single-center cohort comprised of 325 patients with PBC. It was conducted between March 2010 and November 2020 at Nara Medical University Hospital. Patients were diagnosed with PBC according to the Japanese version of the clinical practice guidelines for PBC, developed in 2012 and revised by the Intractable Hepatobiliary Disease Study Group in 2017 with the support of the Ministry of Health, Labor, and Welfare of Japan [15]. The criteria for diagnosis were as follows: (i) positive for anti-mitochondrial antibody (AMA) or AMA-M2, (ii) elevated serum alkaline phosphatase (ALP) and γ-glutamyl transpeptidase (γ-GTP) levels for more than six months, and (iii) presence of typical histological features of PBC on liver biopsy. In this study, 77 patients were treatment-naive when they underwent liver biopsy. All patients were diagnosed with PBC based on serological and histological criteria. Data were collected at baseline and upon treatment with UDCA at a daily dose of 13–15 mg/kg of body weight. The complications of PBC include pruritus, jaundice, ascites, esophageal varices, and hepatocellular carcinoma [16,17]. In this study, patients with PBC were treated with UDCA at a daily dose of 13–15 mg/kg at baseline. The exclusion criteria were as follows: (i) presence of an autoimmune overlap syndrome [18], (ii) use of immunosuppressant medications such as prednisone and azathioprine, and (iii) clinical findings suggestive of concomitant liver disease (i.e., hepatitis B and C virus infections, alcohol-related liver disease, and non-alcoholic fatty liver disease) with a follow-up period within one year. This study was conducted in accordance with the standards of the Declaration of Helsinki and written informed consent was obtained from all patients before enrollment. The Ethics Committee of Nara Medical University Hospital approved the study on 15 March 2009 (No. C2-135 approved).

### 3.2. Histological Evaluation of Liver Tissues According to the Scheuer and Nakanuma Classifications

Liver specimens were obtained from 77 consecutive patients who underwent liver biopsy using a 16-G needle under ultrasound guidance. Tissue sections were stained with hematoxylin and eosin (H&E) and Mallory’s azan stain. The histological stage was evaluated based on both the Scheuer [19] and Nakanuma [20] classifications. The Nakanuma classification was used to determine staging for bile duct loss (BDL) and fibrosis. BDL and fibrosis scores range from 0 to 3, with the corresponding sum of these scores interpreted as follows: 0 is classified as stage 1 (no progression), 1 or 2 as stage 2 (mild progression), 3 or 4 as stage 3 (moderate progression), and 5 or 6 as stage 4 (advanced progression). All liver specimens were confirmed with H&E staining to contain at least fifteen full-portal triads composed of hepatic artery and portal vein branches and the bile duct [21] Prof. Dr. Chiho Obayashi and Dr. Tadashi Namisaki independently reviewed all participants to validate histological characteristics of PBC.

### 3.3. IP Markers

For the non-invasive evaluation of IP, the serum levels of sCD163 were determined using a kit (Cusabio Technology, Houston, TX, USA) [22], while sMR was measured using an enzyme-linked immunosorbent assay (ELISA) kit (LifeSpan BioSciences, Seattle, WA, USA) [23]. Zonulin was measured using a human zonulin ELISA kit (Elabscience, Wuhan, China) [24]. All assays were conducted according to the manufacturer’s instructions. The serum samples were collected at baseline and stored at −80 °C until analyses.

### 3.4. Statistical Analysis

Albumin (ALB), platelet, ALP, total bilirubin (T-Bil), sCD163, sMR, and zonulin levels were normally distributed. AST, ALT, and γ-GTP levels were not normally distributed. Descriptive results for continuous variables were expressed as a mean ± standard deviation and median (interquartile range (IQR)), respectively, whereas categorical variables were presented in a contingency table. Baseline characteristics were compared between the groups using the Mann–Whitney U test for non-normally distributed variables, Student’s t-test for normally distributed variables, and Fisher’s exact test for categorical variables. Spearman’s rank correlation coefficient was used to evaluate the association between histological stages and IP markers. All statistical analyses were conducted using GraphPad Prism version 8.0 for Windows (GraphPad Software, La Jolla, CA, USA). A decision tree model was used to determine the optimal cutoff value of sCD163, sMR, and a combination of both parameters. The prediction accuracy indices of biomarkers for complications, i.e., sensitivity, specificity, and area under the curve (AUC), were evaluated using the receiver operating characteristic curve. The estimation of the confidence interval of AUC and statistical comparison between AUCs were performed using the bootstrap method. Akaike’s information criterion was used to assess the relative quality of a statistical model for a given set of data. Multiple regression analysis was used to identify predictive markers for symptom development. Complication analyses were performed using the Kaplan–Meier method. The log-rank test was used to compare the incidence of complications between two groups. All tests were two-tailed, and a p value of <0.05 was considered statistically significant [25]. The level of significance of Spearman’s rank test was determined as *p* < 0.05 and R > 0.2.

## 4. Conclusions

The main finding of our study was that sMR could predict development of any of these symptoms in patients with PBC. Notably, the greatest predictive power is when sMR is considered in combination with CD163 (AUC = 0.74). Both sMR and sCD163 were correlated with the Nakanuma fibrosis score in the study population. Previously, Bossen et al. have shown that these biomarkers with an OR of as high as 1 poorly estimate the severity of PBC defined by Model for End-Stage Liver Disease (MELD) and Child–Pugh score [14]. Previous reports state that sCD163 increases with the severity of cirrhosis and alongside the degree of portal hypertension [26], whereas serum sMR levels predict portal hypertension and disease severity in patients with alcoholic cirrhosis [27]. Levels of macrophage activation markers have been reported to increase in early-stage liver, diseases including PBC [14,28], and were lower than those in patients with acute liver injury and decompensated cirrhosis [29]. The reason for the difference in serum levels of macrophage activation markers remains unclear; it may be partially accounted for by the fact that the serum levels of these markers could be affected by the presence of background liver disease and the severity of chronic liver disease. These findings suggest that macrophage activation markers increase even in the early stages of PBC.

Hepatic macrophages play a central role in the pathogenesis of PBC [30], including inflammation and fibrosis [31,32]. Their activation increases receptor shedding, resulting in the release of sCD163 and sMR into the circulation system in response to lipopolysaccharide (LPS) stimulation. Activated macrophages shed sCD163 into the circulation after toll-like receptor activation in the presence of LPS in portal blood due to increased IP in patients with compensated advanced liver diseases [33]. On the other hand, CD163 is released in response to direct LPS stimulation, while LPS-mediated shedding of MR results from indirect signaling [33]. LPS has different effects during the release of inflammatory mediators (including sCD163 and sMR) from macrophages because of differences in macrophage polarization states [34]. Furthermore, CD163 and MR on the surface of macrophages are differentially modulated. IL-10 stimulation increases CD163 expression on macrophages, whereas IL-4 and IL-13 promote MR expression [35]. These findings suggest that sCD163 and sMR are involved in distinct inflammatory processes in the gut–liver axis.

Patients with autoimmune and inflammatory diseases, including PBC, are described as having a defective epithelial barrier and increased IP. A possible pathogenic role of bacterial translocation has been reported in PBC. Multiple sequencing studies have been implemented to precisely characterize the microbiota composition of patients with chronic cholestatic liver diseases at both at the fecal and mucosal level. A deficiency in IgA secretion is involved with the penetration of particles in the epithelium through the transcellular pathway. The number of patients with PBC having an altered IP would support the proposed relationship of the mucosal IgA defect in PBC to the disruption of intestinal barrier function [36], which further emphasizes the role of the gut–liver axis in PBC. These findings reinforce the fact that IP markers can estimate the risk for complications and histological progression of PBC.

Nevertheless, this study has several limitations. First, the number of symptomatic patients with PBC was small, because UDCA starts immediately after the diagnosis. Second, the lactulose mannitol test was not performed in this study, although the lactulose mannitol ratio represents an established marker of IP. Third, there were a small number of participants, especially of those with Scheuer stage 4 and a Nakanuma fibrosis score of 3. Future prospective studies are required to validate the findings of the present study.

Collectively, in these cohorts of patients with PBC, sMR plus CD163 showed better predictive power than either marker alone, while the addition of sCD163 did not improve the predictive power of sMR. The dysregulation of macrophage activation is pathogenically associated with various inflammatory and immune disorders. This may be due to the nature of these markers for macrophage activation. More specific biomarkers could be explored in future research for the non-invasive estimation of the risk of developing PBC-related complications.

## Figures and Tables

**Figure 1 ijms-23-09814-f001:**
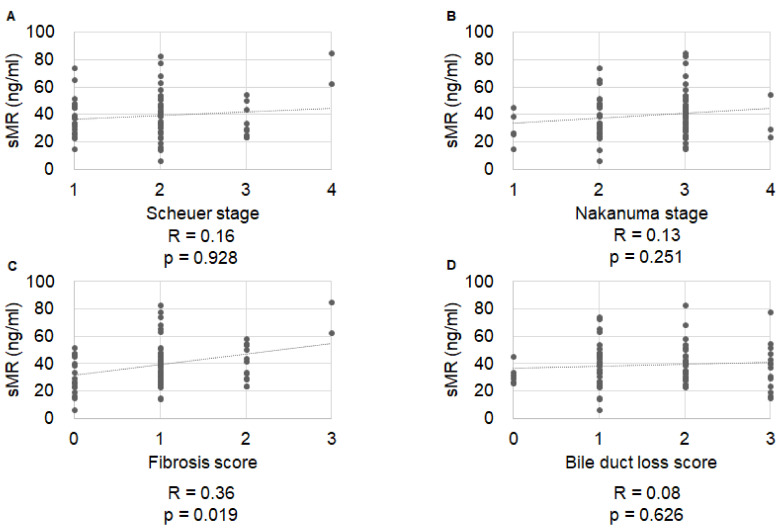
Correlation of soluble mannose receptor with histological stage. (**A**) Scheuer stage, R = 0.16, *p =* 0.928; (**B**) Nakanuma stage, R = 0.13, *p =* 0.251; (**C**) Nakanuma fibrosis score, R = 0.36, *p =* 0.019; (**D**) Nakanuma bile duct score, R = 0.08, *p =* 0.626.

**Figure 2 ijms-23-09814-f002:**
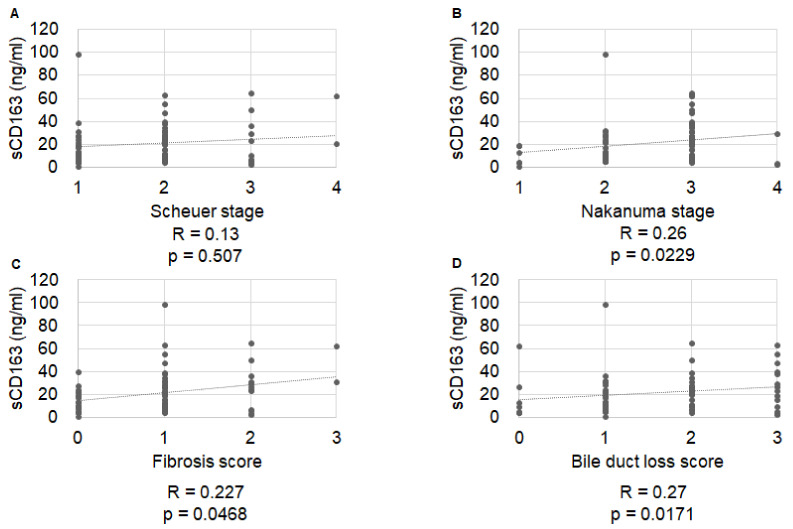
Correlation of serum CD163 levels with histological stage. (**A**) Scheuer stage, R = 0.13, *p =* 0.507; (**B**) Nakanuma stage, R = 0.26, *p =* 0.0229; (**C**) Nakanuma fibrosis score, R = 0.227 *p =* 0.0468; (**D**) Nakanuma bile duct score, R = 0.27 *p =* 0.0171.

**Figure 3 ijms-23-09814-f003:**
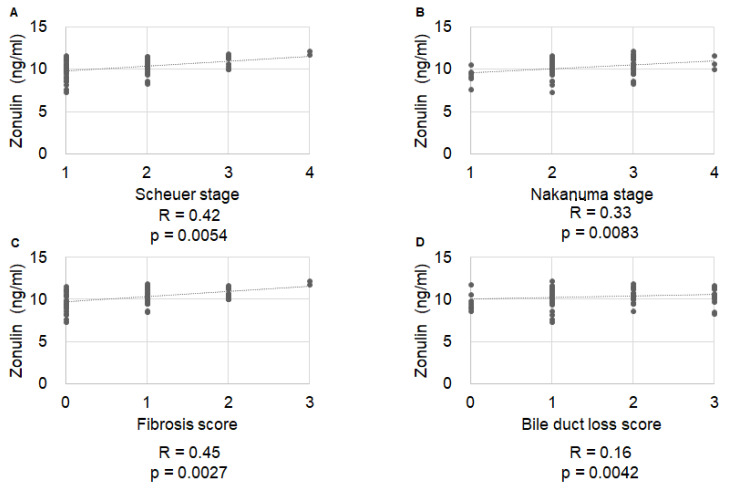
Correlation of serum zonulin levels with histological stage. (**A**) Scheuer stage, R = 0.42, *p =* 0.0054; (**B**) Nakanuma stage, R = 0.33, *p =* 0.0083; (**C**) Nakanuma fibrosis score, R = 0.45, *p =* 0.0027; (**D**) Nakanuma bile duct score, R = 0.16, *p =* 0.0042.

**Figure 4 ijms-23-09814-f004:**
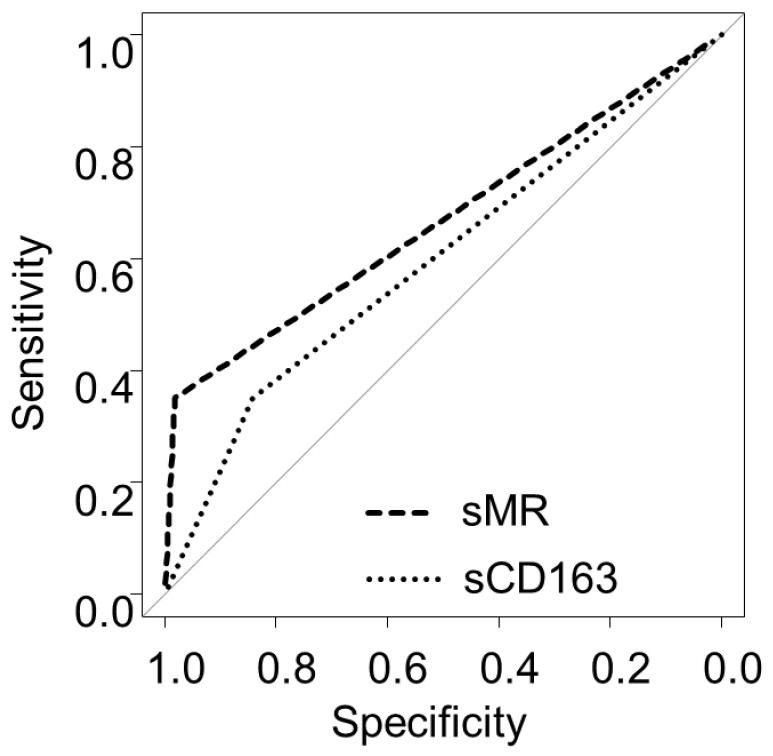
Receiver operating characteristic curves for soluble mannose receptor and soluble CD163 alone and combined.

**Figure 5 ijms-23-09814-f005:**
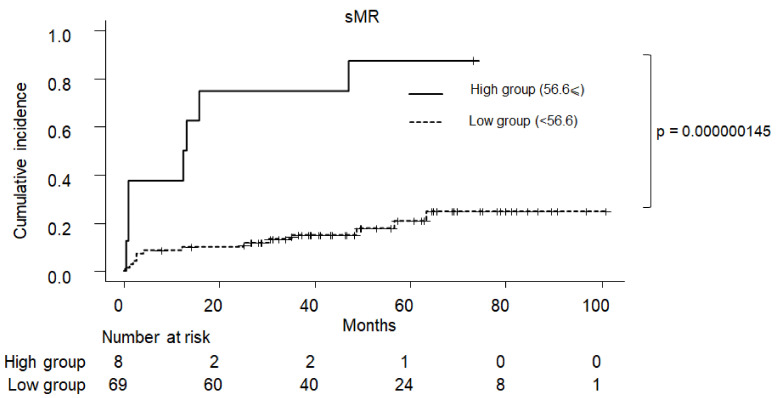
The cumulative incidence of PBC-related complications according to sMR levels. Patients with PBC having a sMR of ≥56.6 developed liver-related complications more frequently than those with sMR of <56.6 (*p* < 0.01). PBC: primary biliary cholangitis, sMR: soluble mannose receptor.

**Figure 6 ijms-23-09814-f006:**
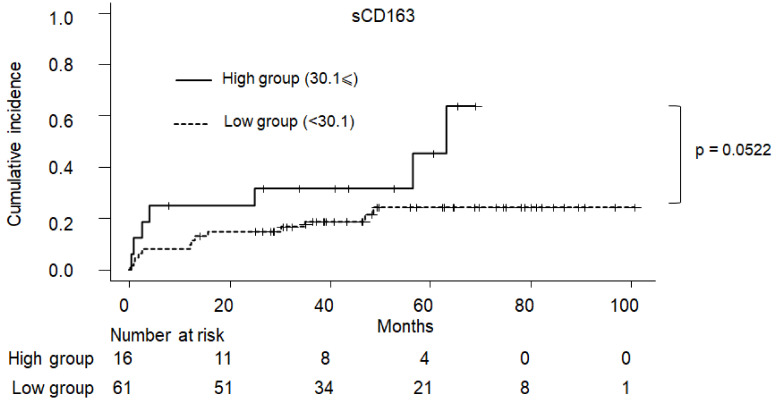
The cumulative incidence of PBC-related complications according to CD163. Patients with PBC having sCD163 of ≥30.1 tended to have more liver complications more frequently than those with sCD163 of <30.1 (*p =* 0.0502). PBC: primary biliary cholangitis.

**Table 1 ijms-23-09814-t001:** Clinical characteristics of patients with primary biliary cholangitis.

Variable	Total (*n* = 77)	No Symptom Development (*n* = 57)	Symptom Development (*n* = 20)	*p* Value
Gender (M/F)	11/66	9/48	2/18	0.718
Age (years) ^†^	63.5 ± 9.8	62.6 ± 9.5	66.1 ± 10.4	0.166
Pruritus/ascites/esophageal varices/HCC/jaundice	12/2/6/2/2	-	12/2/6/2/2	-
Scheuer stage (I/II/III/IV)	25/39/11/2	20/28/9/0	5/11/2/2	0.085
Nakanuma stage (1/2/3/4)	6/28/40/3	5/22/27/3	1/6/13/0	0.474
Fibrosis score (0/1/2/3)	19/43/13/2	16/32/9/0	3/11/4/2	0.076
Bile duct loss score (0/1/2/3)	7/32/24/14	5/23/17/12	2/9/7/2	0.747
Observation period (years) ^†^	4.4 ± 1.8	4.3 ± 1.8	4.5 ± 1.8	0.717
PLT (×10^4^/mL) ^†^	21.2 ± 6.9	22.1 ± 6.0	19.1 ± 7.8	0.087
ALB (g/dL) ^†^	4.1 ± 0.5	4.2 ± 0.4	4.0 ± 0.5	0.262
AST (IU/L) ^‡^	47.0 [32.0–77.0]	46.0 [31.0–63.0]	61.0 [35.0–83.5]	0.149
ALT (IU/L) ^‡^	41.0 [27.0–73.0]	40.0 [27.0–67.0]	53.0 [28.5–75.0]	0.414
ALP (IU/L) ^†^	534.0 ± 369.3	501.2 ± 372.5	627.4 ± 352.6	0.191
γ-GTP (IU/l) ^‡^	131.0 [86.0–261.0]	119.0 [79.0–236.0]	176.0 [117.3–330.5]	0.384
T-Bil (mg/dL) ^†^	1.0 ± 0.7	0.9 ± 0.4	1.3 ± 1.2	0.103
Smr (ng/mL) ^†^	39.5 ± 19.0	36.3 ± 10.9	48.7 ± 31.2	0.011
sCD163 (ng/mL) ^†^	21.2 ± 17.6	18.8 ± 14.9	27.8 ± 22.8	0.048
Zonulin (ng/mL) ^†^	10.3 ± 1.0	10.1 ± 1.3	10.2 ± 1.2	0.759

Categorical variables are presented as a number and (percentage). ^†^ mean ± SD, ^‡^ median (IQR). Normal ranges: platelets (PLT), hepatocellular carcinoma (HCC), 13.1–36.9 × 104/mL; albumin (ALB), 3.9–4.9 g/L; aspartate aminotransferase (AST), 9–37 IU/L; alanine aminotransferase (ALT), 3–43 IU/L; alkaline phosphatase (ALP), 104–328 IU/L; γglutamyl transpeptidase (GGT), 6–52 IU/L; total bilirubin (T-BIL), 0.1–1.1 mg/dL; sMR, soluble mannose receptor; sCD163, soluble CD163.

**Table 2 ijms-23-09814-t002:** Diagnostic accuracy of intestinal permeability markers for symptom development in patients with primary biliary cholangitis.

Variable	Cutoff	Sensitivity	Specificity	PPV	NPV	AUC (95% Confidence Interval)
**sCD163**	30.1	0.35	0.842	0.438	0.787	0.596 (0.479–0.713)
**sMR**	56.6	0.35	0.982	0.875	0.812	0.666 (0.566–0.775)
**sMR + sCD163**	0.254	0.60	0.825	0.545	0.855	0.739 (0.609–0.861)

sMR, soluble mannose receptor; sCD163, soluble CD163; AUC, Area Under Curve; PPV, Positive Predictive Value; NPV, Negative Predictive Value.

**Table 3 ijms-23-09814-t003:** Univariate and multivariate analyses of predictive markers for symptom development in patients with primary biliary cholangitis.

	Univariate Analysis		Multivariate Analysis	
**Variables**	**Odds Ratio (95% CI)**	***p* Value**	**Odds Ratio (95% CI)**	***p* Value**
**56.6 ⩽ Soluble MR**	30.20 (3.410–267.0)	0.00220	33.70 (3.6600–311.0)	0.0019
**30.1 ⩽ sCD163**	2.870 (0.898–9.18)	0.07530	3.410 (0.9310–12.50)	0.0640

## Data Availability

Raw data were generated at Nara University Hospital. The datasets generated during and/or analyzed during the current study are available from the corresponding author (T.N.) on reasonable request.

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
