# Peer review of "Macrophage Activation Markers Predict Liver-Related Complications in Primary Biliary Cholangitis"

_ijms, 2022, doi:10.3390/ijms23179814_

Round 1
Reviewer 1 Report
It is generally accepted that PBC is an autoimmune cholestatic liver diseases with inflammation and bile acids retention. Taking into account that macrophages are key components of the innate immune system, which play important roles in the regulation of the intestinal permeability, the authors measured the serum levels of macrophage activation markers, soluble CD163 (sCD163), soluble mannose receptor (sMR), and zonulin, and their relationship with the development of complications in patients with PBC. The authors undertook these studies despite the fact that AASLD Practice Guidance on PBC found no significant correlation between diseases symptoms and the disease stage. They denote that symptomatic group had significantly higher serum sMR and sCD163 levels than asymptomatic group, and that serum sMR was an independent factor for developing complications on univariate and multivariate analyses. The main finding of these study was that sMR could predict development of clinical complications in patients with PBC.
1/ In the Introduction: Authors should explain what symptoms they mean, dividing patients into symptomatic and asymptomatic.
2/ On the other hand, they should explain for the purpose of the work and in the patient characterization what specific complications are included. It can only be suspected that symptomatic patients are pruritus based on table 2 (20 symptomatic) and the characteristics (20 with a pruritus).
3/ Why is pruritus listed as a complication and not an essential symptom of the disease?
4/ The authors cite complications together with literature references. And which of them were included in the work? (“The complications of PBC include pruritus, jaundice, ascites,
esophageal varices, and hepatocellular carcinoma [15,16].”).
5/ But why was the most common clinical symptom of fatigue not included?
6/ What does it mean: “Out of 77 patients, 20 (26.0%) developed pruritus (n = 12),…”? So how many patients were there with the pruritus: 20 or 12?
7/ The authors should comment on the differences in the histology scores on the Nakanuma scale with the Scheuer scale presented in Table 2. In symptomatic and asymptomatic patients, the most frequent is Nakanuma stage 3 but Sheuer stage 2. Both scales have 4 levels. If the Nakanuma fibrosis score is more useful than the Scheuer stage in terms of the estimation of liver fibrosis progression as Scheuer's classification is characterized by both fibrosis and bile duct changes, why is Fibrosis score 1 the most common?
8/ The authors present the diagnostic accuracy of intestinal permeability markers for symptom development in patients with primary biliary cholangitis – which symptoms?
9/ The authors denote that serum sMR was an independent factor for developing complications- which complications?
10/ Authors should indicate the origin of the cutt-off values (Table 2).
11/ There are many ways to perform multivariate analysis depending on your goals. Which ways authors used? (Cluster Analysis, Correspondence Analysis / Multiple Correspondence Analysis, Factor Analysis, Generalized Procrustean Analysis, MANOVA, Multidimensional Scaling or Multiple Regression Analysis).
12/ In the Table 3, first column, Variables. Is it correct? And it shouldn't be: Soluble SM≥56.6?
Author Response
26 th August 2022
Dear Editor-in-Chief, Associate Editor, and Referees:
Please find enclosed our edited manuscript in Word format.
Title: Macrophage activation markers predict liver-related complications in primary biliary cholangitis
Authors: Yukihisa Fujinaga1, Tadashi Namisaki1, Yuki Tsuji1, Junya Suzuki1, Koji Murata1, Soichi Takeda1, Hiroaki Takaya1, Takashi Inoue2, Ryuichi Noguchi1, Yuki Fujimoto1, Masahide Enomoto1, Norihisa Nishimura1, Koh Kitagawa1, Kosuke Kaji1, Hideto Kawaratani1, Takemi Akahane1, Akira Mitoro1, Hitoshi Yoshiji1i
Name of Journal: IJMS
Manuscript No: ijms-1886836.R1
We are very grateful to you and the reviewer for the comments and thoughtful suggestions regarding our manuscript. We have thoroughly addressed all concerns and issues raised and have revised our manuscript accordingly. All changes in the revised version are highlighted in yellow. We believe that the manuscript has been greatly improved and hope it has reached the standards of Hepatology Research. Once again, we acknowledge your comments that have been extremely valuable in improving the quality of our manuscript. We have provided point-by-point responses to the reviewers’ comments below.
It is generally accepted that PBC is an autoimmune cholestatic liver diseases with inflammation and bile acids retention. Taking into account that macrophages are key components of the innate immune system, which play important roles in the regulation of the intestinal permeability, the authors measured the serum levels of macrophage activation markers, soluble CD163 (sCD163), soluble mannose receptor (sMR), and zonulin, and their relationship with the development of complications in patients with PBC. The authors undertook these studies despite the fact that AASLD Practice Guidance on PBC found no significant correlation between diseases symptoms and the disease stage. They denote that symptomatic group had significantly higher serum sMR and sCD163 levels than asymptomatic group, and that serum sMR was an independent factor for developing complications on univariate and multivariate analyses. The main finding of these study was that sMR could predict development of clinical complications in patients with PBC.
1/ In the Introduction: Authors should explain what symptoms they mean, dividing patients into symptomatic and asymptomatic.
Authors Response
Thank you for your valuable comments. No patients developed any symptoms at the beginning of the observation period. Among the 20 (26.0%) out of the 77 patients overall, 12 developed pruritus, 2 ascites, 6 esophageal varices, 2 hepatocellular carcinoma, and 2 jaundice. Some patients developed several symptoms. 57 (84.0%) patients remained symptomatic throughout the study period. We have accordingly included a description explaining these findings on page 2, lines 9–15.
2/ On the other hand, they should explain for the purpose of the work and in the patient characterization what specific complications are included. It can only be suspected that symptomatic patients are pruritus based on table 2 (20 symptomatic) and the characteristics (20 with a pruritus).
Authors Response
Thank you for your valuable comments. We aimed to identify biomarkers to predict the development of clinical symptoms in patients with PBC. No patients developed any symptoms at the beginning of the observation period. Among 20 (26.0%) out of the 77 patients overall, 12 developed pruritus, 2 developed ascites, 6 developed esophageal varices, 2 developed hepatocellular carcinoma and 2 developed jaundice. We have accordingly included a description explaining these findings on page 2, lines 8–11 and 28-30.
3/ Why is pruritus listed as a complication and not an essential symptom of the disease?
Authors Response
Thank you for your valuable comments. The most common symptom of PBC requiring treatment is pruritus in patients with PBC (Tajiri K et al J gastroenterol 2017, new reference no. 9). Pruritus severity is unrelated to the severity of the liver disease. We have accordingly included a description explaining these findings on page 2, lines 14-15.
4/ The authors cite complications together with literature references. And which of them were included in the work? (“The complications of PBC include pruritus, jaundice, ascites,
esophageal varices, and hepatocellular carcinoma [15,16].”).
Authors Response
Thank you for your valuable comments. The complications of PBC include pruritus, jaundice, ascites, esophageal varices, and hepatocellular carcinoma [16,17]. All of them are included in this study. We have accordingly included a description explaining these findings on page 2, lines 45-46.
5/ But why was the most common clinical symptom of fatigue not included?
Authors Response
Thank you for your valuable comments. The most common symptom of PBC requiring treatment is pruritus in patients with PBC (Tajiri K et al J gastroenterol 2017, new reference no 9). No patients had fatigue in this study. We have accordingly included a description explaining these findings on page 2, lines 14-16.
6/ What does it mean: “Out of 77 patients, 20 (26.0%) developed pruritus (n = 12),…”? So how many patients were there with the pruritus: 20 or 12?
Authors Response
Thank you for your valuable comments. Among the 20 (26.0%) out of the 77 patients overall, 12 developed pruritus, 2 ascites, 6 esophageal varices, 2 hepatocellular carcinoma, and 2 jaundice. Some patients developed several symptoms. 57 (84.0%) patients remained symptomatic throughout the study period. We have accordingly included a description explaining these findings on page 2, lines 10–13.
7/ The authors should comment on the differences in the histology scores on the Nakanuma scale with the Scheuer scale presented in Table 2. In symptomatic and asymptomatic patients, the most frequent is Nakanuma stage 3 but Sheuer stage 2. Both scales have 4 levels. If the Nakanuma fibrosis score is more useful than the Scheuer stage in terms of the estimation of liver fibrosis progression as Scheuer's classification is characterized by both fibrosis and bile duct changes, why is Fibrosis score 1 the most common?
Authors Response
Thank you for your valuable comments. Notably, patients with early-stage PBC treated with ursodeoxycholic acid (UDCA) have comparable prognosis with age-matched healthy controls (Kawata K et al, Hepatol. Res 2017, new reference No 2). UDCA therapy delays the progression of liver fibrosis in PBC. Hence, more patients with early-stage PBC are now being treated with UDCA (Tanaka K et al, J Gastroenterol 2016, new reference No 3). We have accordingly included a description explaining these findings on page 1, line 54– page 2, line 3.
8/ The authors present the diagnostic accuracy of intestinal permeability markers for symptom development in patients with primary biliary cholangitis – which symptoms?
Authors Response
Thank you for your valuable comments. Intestinal permeability markers could predict development of any of these symptoms in patients with PBC.
9/ The authors denote that serum sMR was an independent factor for developing complications- which complications?
Authors Response
Thank you for your valuable comments. serum sMR can could predict development of any of these symptoms in patients with PBC. We have accordingly included a description explaining these findings on page 10, lines 1–2.
10/ Authors should indicate the origin of the cutt-off values (Table 2).
Authors Response
Thank you for your valuable comments. A decision tree model was used to determine the optimal cutoff value of sCD163, sMR, and a combination of both parameters. We have accordingly included a description explaining these findings on page 3, lines 36-37.
11/ There are many ways to perform multivariate analysis depending on your goals. Which ways authors used? (Cluster Analysis, Correspondence Analysis / Multiple Correspondence Analysis, Factor Analysis, Generalized Procrustean Analysis, MANOVA, Multidimensional Scaling or Multiple Regression Analysis).
Authors Response
Thank you for your valuable comments. Multiple Regression Analysis was used to identify predictive markers for symptom development. We have accordingly included a description explaining these findings on page 3, lines 42–43.
12/ In the Table 3, first column, Variables. Is it correct? And it shouldn't be: Soluble SM≥56.6?
Authors Response
Thank you for pointing this out. 56.6⩽ sMR is correct. We have revised the manuscripts accordingly.
Reviewer 2 Report
This is an interesting study from Fujinaga and colleagues, where the authors support the use of macrophage activation markers for predicting liver-related complications in PBC.
The title is accurate and reflects the context.
Abstract is satisfactory, summarizing the results and conclusions.
Introduction is satisfactory but, I should make the following comments: 1) all patients diagnosed with PBC are treated with UDCA irrespective of the stage of the disease and variable prognostic scores evaluate the treatment response and the risk for complications on 6 and most of them at 12 months of treatment initiation. So, the aim of this study is to find markers that could predict patients at risk for complications and/or unfavourable outcome earlier than this period of time or patients with advanced risk for complications even when they present good response to treatment with UDCA?, 2) the role of intestinal permeability in PBC as well as the reason for which these two markers were chosen should be analyzed in more details.
Patients and Methods is well organized and presented. A minor comment refers to the diagnosis of PBC as it seems that only patients with AMA positive were selected, as ANA autoantibodies-PBC specific are not mentioned.
As far as the results part is concerned, I would suggest the first part "3.1 Baseline clinical characteristics" as well as Table 1 to be reorganized so as the number of patients that presented complications, the kind of complications as well as their distribution in histological stages according to different scores to be more comprehensive by the reader.
Moreover, another issue is that in the results part symptoms of the disease such as pruritus and liver related complications such as ascites are analyzed together. It should be clear what these two markers are used for. Prediction of development of symptoms, or prediction of advanced stage liver disease, as it is well known that symptomatic patients (patients with pruritus or fatigue) have not necessary advanced histological stages.
Pictures and Tables are necessary and satisfactory (see comment about table 1).
English are acceptable.
Discussion is well organized and clearly states the limitations of the study.
References are satisfactory.
In conclusion it is an interesting and well presented work, with easily applicable methodology that opens the way for larger studies to be conducted in the same field.
Author Response
- Patients and Methods is well organized and presented. A minor comment refers to the diagnosis of PBC as it seems that only patients with AMA positive were selected, as ANA autoantibodies-PBC specific are not mentioned.
Authors Response
Thank you for your valuable comments. Antinuclear antibodies (ANA) become positive in approximately 30% to 50% of patients with PBC. However, patients were diagnosed with PBC according to the Japanese version of the clinical practice guidelines for PBC, which was developed in 2012 and revised by the Intractable Hepatobiliary Disease Study Group in 2017, with the support of the Ministry of Health, Labor, and Welfare of Japan [15]. In this current study, Tht criteria for diagnosis were as follows: (i) positive for anti-mitochondrial antibody (AMA) or AMA-M2, (ii) elevated serum alkaline phosphatase (ALP) and γ-glutamyl transpeptidase (γ-GTP) levels for more than 6 months, and (iii) presence of typical histological features of PBC on liver biopsy.
- As far as the results part is concerned, I would suggest the first part "3.1 Baseline clinical characteristics" as well as Table 1 to be reorganized so as the number of patients that presented complications, the kind of complications as well as their distribution in histological stages according to different scores to be more comprehensive by the reader.
Authors Response
Thank you for your valuable comments. We revised Table 1, accordingly.
Moreover, another issue is that in the results part symptoms of the disease such as pruritus and liver related complications such as ascites are analyzed together. It should be clear what these two markers are used for. Prediction of development of symptoms, or prediction of advanced stage liver disease, as it is well known that symptomatic patients (patients with pruritus or fatigue) have not necessary advanced histological stages.
Authors Response
The combination of sMR and sCD163 could predict development of any of these symptoms in patients with PBC. The most common symptom of PBC requiring treatment is pruritus in patients with PBC [Tajiri K et al J gastroenterol 2017, new reference no. 9]. However, no patients had fatigue in this study. Pruritus severity is unrelated to the severity of the liver disease. Therefore, all complications of PBC including pruritus, jaundice, ascites, esophageal varices, and hepatocellular carcinoma are included in this study. We have accordingly included a description explaining these findings on page 2, lines 14–17 and line 43-44.